# Effect of Citrate on the Size and the Magnetic Properties of Primary Fe₃O₄ Nanoparticles and Their Aggregates

Andrea Atrei [1,*], Fariba Fahmideh Mahdizadeh [1], Maria Camilla Baratto [1] and Andrea Scala [2]

1   Dipartimento di Biotecnologie, Chimica e Farmacia, Università di Siena, 53100 Siena, Italy;
    f.fahmidehmahdiza@student.unisi.it (F.F.M.); mariacamilla.baratto@unisi.it (M.C.B.)
2   Dipartimento di Scienze Fisiche, della Terra e dell'Ambiente, Università di Siena, 53100 Siena, Italy;
    andrea.scala@unisi.it
*   Correspondence: atrei@unisi.it

**Abstract:** The size, size distribution and magnetic properties of magnetite nanoparticles (NPs) prepared by co-precipitation without citrate, in the presence of citrate and citrate adsorbed post-synthesis were studied by X-ray Diffraction (XRD), Dynamic Light Scattering (DLS), Electron Paramagnetic Resonance (EPR) and magnetization measurements. The aim of this investigation was to clarify the effect of citrate ions on the size and magnetic properties of magnetite NPs. The size of the primary NPs, as determined by analysing the width of diffraction peaks using various methods, was ca. 10 nm for bare magnetite NPs and with citrate adsorbed post-synthesis, whereas it was around 5 nm for the NPs co-precipitated in the presence of citrate. DLS measurements show that the three types of NPs form aggregates (100–200 nm in diameter) but the dispersions of the citrate-coated NPs are more stable against sedimentation than those of bare NPs. The sizes and size distributions determined by XRD are in good agreement with those of the magnetic domains obtained by fitting of the magnetization vs. magnetic field intensity curves. Magnetization vs. magnetic field intensity curves show that the three kinds of sample are superparamagnetic.

**Keywords:** magnetite; nanoparticles; XRD; DLS; magnetic properties; superparamagnetism



## 1. Introduction

Iron oxide nanoparticles (NPs) are interesting in view of their applications ranging from biomedical field to environmental remediation. Among the several synthetic procedures developed to prepare magnetite (Fe₃O₄) and maghemite (γ-Fe₂O₃) NPs, co-precipitation is one of the most commonly used methods. The various co-precipitation procedures are variants of the Massart method [1]. This preparation procedure, in which magnetite NPs are precipitated by adding a base to aqueous solutions of Fe(II) and Fe(III) salts, is not experimentally demanding since it does not require high temperatures, toxic precursors or solvents. The reaction mechanism leading to the formation of magnetite NPs, involving several iron oxides and oxyhydroxide intermediates, is not completely clarified [2–5]. Reaction conditions, such as the concentration of the reactants, pH, temperature, reaction temperature and time rate of the alkali addition, and the types of counterions of the Fe(II) and Fe(III) salts, determine to a large extent the average size, the size distribution of the individual iron oxide NPs and, thus, their magnetic properties [6–9]. The co-precipitation in aqueous environment leads to the formation of aggregates (which can reach the micrometric size) consisting of individual NPs. The ionic strength and pH of the reaction mixture influence the aggregation process. NPs of magnetite prepared by co-precipitation are negatively charged since the co-precipitation occurs at pH (10–12) well above the point of zero charge of Fe₃O₄ particles [10]. The electrostatic repulsion between the negatively charged NPs is reduced by the relatively high ionic strength of the solution and aggregation of NPs occurs. Hence, the formation of clusters of individual NPs limits the colloidal stability of magnetite NP dispersions and influences the magnetic properties

of these clusters in which the NPs are in close proximity. For most applications the surface of the NPs must be modified in order to prevent or at least reduce their aggregation. A reach literature exists on the surface coating by adsorption of ions and polymers to stabilize dispersions of magnetic NPs in various environments [11–14], just to cite a few reviews. Citrate ions adsorbed at the surface of NPs are used to stabilize aqueous dispersions of metallic [15,16] and metal oxide [17] NPs, including those of iron oxides [18]. Citrate ions are chemisorbed via coordination bonding of one or two carboxylate groups with iron ions at the surface of the NPs. At pH above 4 the negatively charged surface provides electrostatic stabilization, at least at relatively low ionic strength [19–21]. A reliable determination of the size and size distribution of the individual NPs as well as of their aggregates is fundamental for the improvement of the synthetic procedures and to understand the relationships between size and magnetic properties. In the present work, we investigated by means of X-ray Diffraction (XRD), Dynamic Light Scattering (DLS), magnetization measurements and Electron Paramagnetic Resonance (EPR) the size and magnetic properties of magnetite NPs prepared by co-precipitation without and with citrate under various conditions. The aim of this investigation was to clarify the effect of citrate ions on the size of $Fe_3O_4$ NPs and on their magnetic properties. In previous studies it was found that the presence of citrate during the co-precipitation leads to a reduction in the size of the NPs [22,23]. However, there are many aspects of this system to be investigated more deeply. In particular no information about the particle size distribution (PSD) was reported. A study of the particle size, PSD, aggregation of the NPs and their effect on the magnetic properties of magnetite NPs is of fundamental relevance to tailor the preparation procedures in view of the applications of these NPs. In order to clarify the effect of citrate on particle size and PSD, the coating with citrate was carried out by co-precipitation of magnetite NPs in the presence of citrate (in situ) and by adsorbing citrate ions on the NPs after synthesis (ex situ). The size of the individual NPs was determined by analysing the width of the XRD peaks not only employing the well-established Scherrer and Williams-Hall methods but also by using a procedure introduced by Pielaszek [24] and a fitting analysis based on the Rietveld approach [25,26]. The last methods allow one to obtain information about the PSD. In addition to that, the size of the primary NPs prepared under various conditions was estimated by fitting the magnetization vs. the intensity of magnetic field curves [27–29]. The results of the XRD analysis are compared with those of the fitting of the magnetization curves to corroborate the conclusions derived from this study. These methods provide more reliable results than the analysis by Transmission Electron Microscopy (TEM) or Atomic Force Microscopy (AFM) when NPs form aggregates. Due to particle aggregation the size of the individual particles cannot be determined by means of an automated analysis of TEM images and the manual observation is likely to overestimate the particle mean size since the isolated and larger ones are more easy to be seen [30]. DLS was used to determine the size of the aggregates of NPs in the dispersions. Zero Field Cooling/Field Cooling (ZFC/FC) curves and EPR spectra were measured in order to derive information about the interactions between bare and citrate-coated (prepared in both ways) magnetite NPs.

## 2. Materials and Methods

All materials were research grade products (Sigma-Aldrich, St. Louis, MO, USA) and used as received.

### 2.1. Preparation of $Fe_3O_4$ NPs

1.17 g of $FeCl_3 \cdot 6H_2O$ (purity 97%) and 0.93 g of $Fe(NH_4)_2(SO_4)_2 \cdot 6H_2O$ (Mohr salt, purity 99%) were dissolved in 50 mL of double distilled water (DDW), deoxygenated by $N_2$ bubbling. A 10% excess of Mohr salt with respect to the stoichiometric Fe(II)/Fe(III) ratio 1:2 was used to compensate for possible oxidation of Fe(II). The solution was heated at 60 °C and 22 mL of 1 M NaOH (prepared in deoxygenated water) were rapidly added. The reaction mixture was kept for 15 min at this temperature and let to cool to room

temperature (RT) in ca. 1 h under $N_2$ bubbling. The black precipitate of magnetite NPs was separated from the solution by means of a magnet, washed several times with DDW and dried in a flux of $N_2$. Hereafter, the sample prepared in this way is indicated as A.

For the coating of the $Fe_3O_4$ NPs with citrate, 0.5 g of NPs were dispersed in 100 mL of DDW and 1.90 g of sodium citrate dihydrate (purity 99%) were added (citrate ion concentration: 0.065 M) corresponding to 1:1 molar ratio of Fe to citrate. The pH was adjusted to 6 with drops of a HCl solution and the reaction was carried out for 5 h under mechanical stirring. The pH was measured by means of a pH meter (pH700, Eutech Instruments). The NPs were separated magnetically and dried under a $N_2$ flux. This sample of NPs with the citrate coating after the co-precipitation will be referred to as sample B (or ex situ).

Magnetite NPs were co-precipitated by the same procedure as for sample A and B but adding 1.90 g of sodium citrate dihydrate to the Fe(II)/Fe(III) solution (1:1 molar ratio of Fe to citrate). The colour of the iron salts solution changed from orange to pale green upon addition of citrate indicating the formation of complexes with the iron ions. The NPs were separated from the solution by means of a magnet and by centrifugation (2 h at 10,000 rpm, 12,000× *g*). Subsequently the NPs were dried under a $N_2$ flux. Hereafter the NPs precipitated in the presence of citrate are indicated as sample C (or in situ). For samples B and C the coating with citrate ions was monitored by means of Fourier Transform Infrared spectroscopy (Figure S1, Supplementary Materials).

### 2.2. XRD Measurements

A powder X-ray diffractometer (Philips X'Pert PRO PW 3040/60), working in the Bragg-Brentano geometry, equipped with a X'CeleratorPW3015 detector, a multiple purpose sample stage for massive specimens and mono-capillary collimator tube was used. Diffractograms were collected with non-monochromatized Cu $K_\alpha$ radiation (mean wavelength: 0.15418 nm), in the 10–70° 2θ range and with a step of 0.017°. Diffractograms of a silicon sample with micro sized grains were measured to estimate the instrumental contribution to the width of the diffraction peaks. For the analysis of the diffractograms Origin 8.5 software was used. After background subtraction, the peaks in the diffractograms were fitted by using a Voigt function for the determination of the peak widths needed in the Scherrer, Williamson-Hall and Pielaszek methods. The software PM2K, developed by Leoni and Scardi based on the Rietveld method [26] was used to determine the coherence domain sizes and their size distributions.

### 2.3. DLS Measurements

Dynamic light scattering (DLS) and ζ-potential measurements were performed by means of a Zetasizer Nano ZS90 instrument (Malvern, UK). The instrument uses a He/Ne laser (wavelength 633 nm) and the scattering angle is 173°. Hydrodynamic diameters are expressed as Z-average values. PSD curves were obtained by the instrument software using the non-negative least square fitting method. Number distribution curves were calculated on the basis of the Mie theory using the optical properties of bulk magnetite (refractive index, n: 2.3602; absorption coefficient, k: 0.14716). ζ-potential values were determined from the electrophoretic mobility measurements applying the Smoluchowski approximation. NPs were dispersed in DDW. The pH of the aqueous dispersions was ca. 6.5. The concentration of all samples was of the order of 0.05 mg/mL and the measurements were performed at 25 °C. The reported values of the Z-average diameter and of the polydispersity index (PDI) are the average of three sets of acquisitions. The dispersions were sonicated (35 kHz, ultrasonic nominal power 80 W) for 30 min before DLS measurements.

### 2.4. Magnetization Curves

Magnetization versus magnetic field intensity and ZFC/FC curves were measured by using a Superconducting Quantum Interference Device (SQUID) magnetometer (Quantum

Design Ltd. San Diego, CA, USA) operating with a maximum magnetic field of 50,000 Oe in the 2–400 K temperature range. Hysteresis loops were measured cyclically by varying the magnetic field from 50,000 Oe. Magnetization curves at 300 K were acquired in the range from 50,000 to −300 Oe to check for possible remanence. ZFC/FC were measured with a probe field of 50 Oe from 2.5 to 300 K, after having cooled the sample in the absence (ZFC) or in the presence of the probe field (FC). Magnetic domain sizes and their size distribution functions were determined by fitting the magnetization versus magnetic field intensity curves measured at RT. In the fitting process the saturation magnetization and parameters related to the average size and standard deviation of the size distribution functions were varied on a grid of values until the minimum of the sum of the squares of the residuals was achieved. The error bars of the parameters were estimated from the dependence on the changes of the parameters of the sum of the squares of the residuals.

### 2.5. EPR Measurements

The EPR measurements were carried out by using a Bruker ELEXSYS E580 spectrometer at the X-band frequency ($\upsilon$ ~9.8 GHz) with a 41225HGE/0208 cavity. Sample were contained in quartz capillaries. The NPs were mixed and grinded in mortar with a weight ratio NPs:KBr of 1:300 in order to attenuate the intensity of the signal. The spectra were collected at RT using a microwave power of 4.5 mW. EPR spectra were recorded with two modulation amplitudes of 1G and 5G for two scan ranges, 3000–4000 G and 3100–4100 G. The values of the acquisition parameter were chosen on the basis of previous works and optimized to improve the quality of the spectra.

## 3. Results

### 3.1. XRD

The diffractograms measured for the three kinds of sample are shown in Figure 1. The diffraction peaks are indexed on the basis of the structure of magnetite. The lattice parameters derived from these measurements are the same within the experimental accuracy for all the samples (Table 1). Magnetite and maghemite ($\gamma$-$Fe_2O_3$) have the same structure (cubic inverse spinel) and very similar lattice parameters (0.8397 nm and 0.8315 nm, respectively). Hence, it is not possible to discriminate between these two iron oxides, in particular for a core–shell structure with a magnetite core covered by a thin shell of maghemite (which is probably the present case).

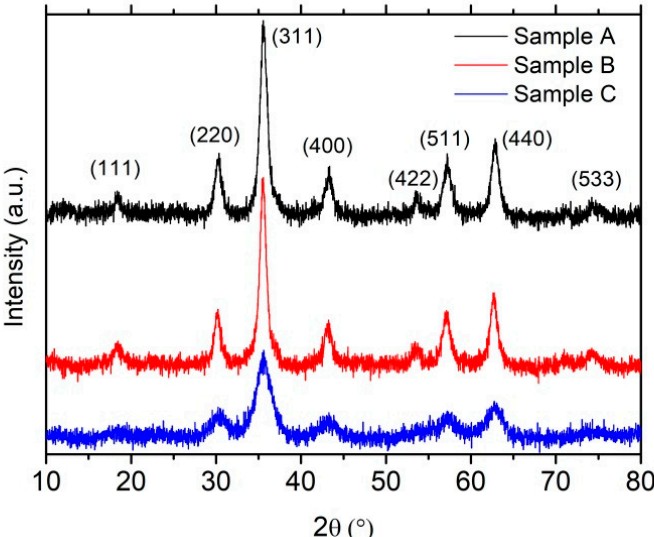

**Figure 1.** Diffractograms measured for sample A (bare $Fe_3O_4$ NPs), sample B (citrate-coated NPs, ex situ) and sample C (citrate-coated NPs, in situ). The curves are offset to improve the clarity of the plot. The main peaks in the diffractograms are indexed on the basis of the structure of magnetite.

**Table 1.** Results of the analysis of the diffractograms by using the various methods. The accuracy on the lattice parameter is $\pm$ 0.001 nm. <D> calculated with all methods is the average weighted by the volume. For Scherrer and Pielaszek methods, the values of <D> are the average of the values derived from the widths of the most intense peaks in the diffractrograms. $\sigma$ is the standard deviation of the PSD.

| Sample A (Bare Fe$_3$O$_4$ NPs) | $a_0$ (nm) | <D> (nm) | $\sigma$ (nm) |
|---|---|---|---|
| Scherrer | 0.837 | 8.4 $\pm$ 0.3 | - |
| Williamson-Hall | - | 8.6 $\pm$ 0.7 | - |
| Pielaszek | - | 8.7 $\pm$ 0.5 | 3.8 $\pm$ 0.6 |
| Rietveld (PM2K) | 0.836 | 9.7 $\pm$ 0.1 | 2.4 $\pm$ 0.1 |
| **Sample B (Fe$_3$O$_4$ NPs cit. ex situ)** | | | |
| Scherrer | 0.837 | 9.2 $\pm$ 0.5 | - |
| Williamson-Hall | - | 9.6 $\pm$ 1.5 | - |
| Pielaszek | - | 8.9 $\pm$ 0.8 | 4.1 $\pm$ 0.7 |
| Rietveld (PM2K) | 0.838 | 10.3 $\pm$ 0.7 | 2.6 $\pm$ 0.2 |
| **Sample C (Fe$_3$O$_4$ NPs cit. in situ)** | | | |
| Scherrer | 0.837 | 3.4 $\pm$ 0.3 | - |
| Williamson-Hall | - | 4.0 $\pm$ 1.0 | - |
| Pielaszek | - | 3.7 $\pm$ 0.3 | 1.6 $\pm$ 0.5 |
| Rietveld (PM2K) | 0.838 | 4.6 $\pm$ 0.2 | 0.9 $\pm$ 0.2 |

From the visual comparison of these curves we can see that the peaks of samples A and B have almost the same width whereas those of sample C are broader. The average sizes of the single NPs were estimated by using three different methods in order to be more confident on the obtained results. The basic assumption of the analysis is that each NP consists of a single coherence domain and that the size of the domains is smaller than the coherence length of experimental set-up. When the NPs form aggregates (such as in the present case) a reliable determination of the average size of the individual (primary) NPs is awkward by analysing TEM images (Figure S2, Supplementary Materials) [30]. The results of the analysis are summarized in Table 1. The Scherrer method allows one to estimate the average size of the particles from the width of the diffraction peaks. NPs can be considered spherical (see Figure S2, Supplementary Materials) and the size of the NPs identified with their diameter. The average diameter reported in Table 1 for each kind of sample is the average of the values calculated for the peaks in the diffractogram. In this way it is possible to compensate for possible errors in the measurement of the peak width and to the direction dependence of the size of the crystallites. The results of the analysis by the Scherrer method confirm that the NPs in sample C have a smaller size than those in samples A and B. The Scherrer method attributes entirely the width of a diffraction peak to the limited size of the crystal grains. However, an additional contribution to the broadness of a peak may come from the microstrain induced by crystal defects (dislocations, for instance). This contribution was estimated by analysing the data on the basis of the Williamson-Hall method. The data for all the three kinds of sample follow only approximately the linear behaviour predicted by the Williamson-Hall method (Figure S3 Supplementary Materials) [31]. This may be due to deviations from the uniform strain model assumed in this version of Williamson-Hall method [32]. However, the analysis indicates that the main contribution to the width of the diffraction peaks comes from the nanometric size of the crystals. Hence, the results of the Williamson-Hall approach for the size of the particles are fully consistent with those of Scherrer analysis (see Table 1). In addition to the average diameter also the PSD curves were determined by analysing the diffraction peak profiles. The average diameters of the NPs and the standard deviations, $\sigma$, for the three kinds of sample calculated by using the Pielaszek and the Rietveld methods are reported in Table 1. The PSD curves of samples A, B and C are shown in Figure 2. The PSD function used in the Pielaszek method is the gamma distribution function. This function,

which is convenient from the computational point of view, is a good approximation of the lognormal distribution curve, as shown in Figure 2, dashed lines. The best fit of the diffractograms of the three samples by means of the Rietveld method implemented in the PM2K software are shown in Figure S4, Supplementary Materials. The average values of the relevant parameters determined by all the methods are reported in Table 2. The analysis of the XRD data clearly show that the average diameter of magnetite.

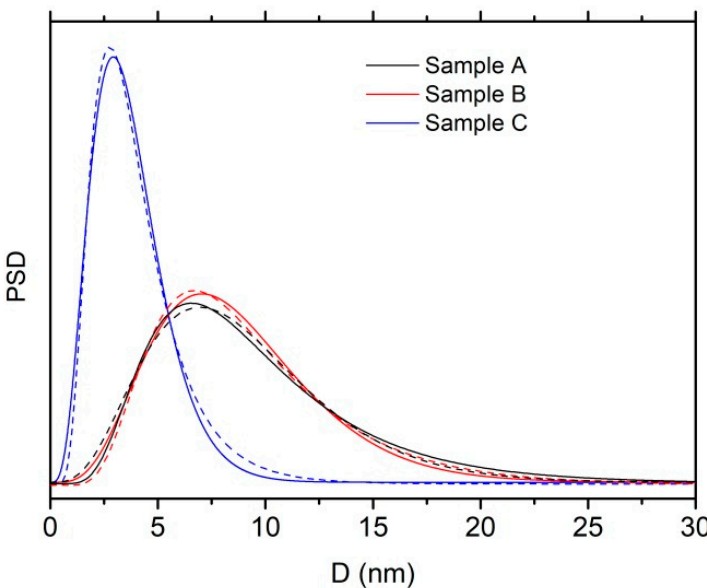

**Figure 2.** Solid line: domain size distributions of sample A, B and C obtained by the Pielaszek method. Dashed lines: fitting of the Gamma distribution functions with lognormal distributions.

**Table 2.** For samples A, B and C, the averages of <D> determined by the Scherrer, Williamson-Hall, Pielaszek and Rietveld methods, the averages of $\sigma$ obtained by the Pielaszek and Rietveld procedures and the relative standard deviations, $\sigma/<D>$, are also shown. The averages of the $a_0$ values derived by using the Scherrer and Rietveld methods are reported.

|  | <D> (nm) | $\sigma$ (nm) | $\sigma/<D>$ | $a_0$ (nm) |
|---|---|---|---|---|
| Bare NPs (sample A) | $8.8 \pm 0.7$ | $3.1 \pm 0.4$ | $0.35 \pm 0.07$ | $0.837 \pm 0.001$ |
| Sample B ($Fe_3O_4$ NPs cit. ex situ) | $9.5 \pm 0.9$ | $3.3 \pm 0.4$ | $0.35 \pm 0.07$ | $0.837 \pm 0.001$ |
| Sample C ($Fe_3O_4$ NPs cit. in situ) | $3.9 \pm 0.5$ | $1.2 \pm 0.4$ | $0.31 \pm 0.14$ | $0.837 \pm 0.001$ |

NPs co-precipitated in the presence of citrate (sample C) is smaller than those of samples A and B. On the other hand, the size of bare NPs is not significantly different from that of the ex situ NPs. This result suggests than when magnetite NPs are immersed in the citrate solution at pH 6 there is not dissolution of the oxide. The standard deviations of sample A and B are larger than that of sample C but the relative standard deviations (standard deviation divided by the average diameter) are barely the same. Hence, we can conclude that the presence of citrate during co-precipitation of magnetite leads to smaller primary NPs than those prepared in absence of citrate but with the same level of dispersion.

### 3.2. DLS

Intensity of scattered light vs. the hydrodynamic diameter curves measured for aqueous dispersions of the three kinds of sample are shown in Figure 3 and the results of the measurements are summarized in Table 3.

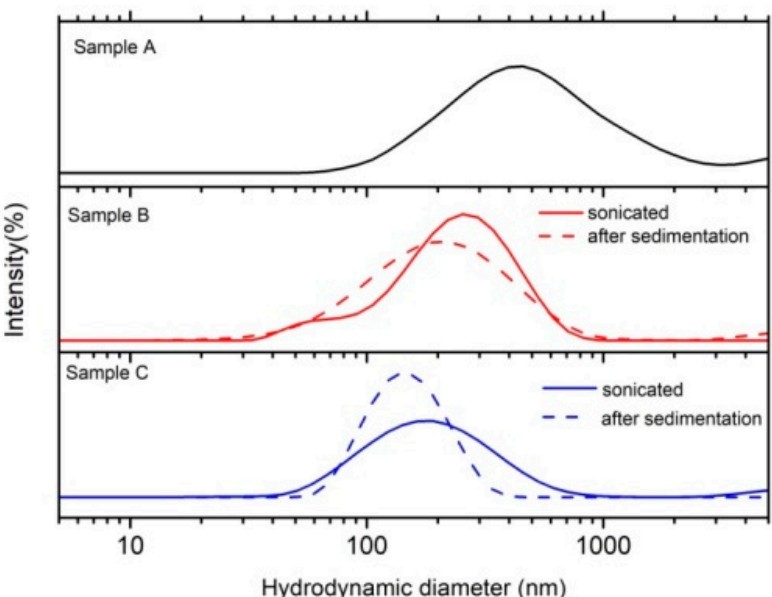

**Figure 3.** Scattered light intensity vs. hydrodynamic diameter curves measured for the samples A, B and C immediately after sonication (solid lines) and after one week (dashed lines).

**Table 3.** Results of the DLS and $\zeta$-potential measurements for the three kinds of samples. The hydrodynamic diameter (nm), the PDI and the $\zeta$-potential (mV) are reported. For the bare NPs one week after preparation DLS measurements could not be performed. For the other samples. The concentration of the dispersions was ca. 0.02 mg/mL for all samples and the pH 6.5. Temperature 298 K.

|  | Hydrodynamic Diameter (nm) | PDI | $\zeta$-Potential (mV) |
|---|---|---|---|
| Sample A | $398 \pm 22$ | $0.39 \pm 0.1$ | $16 \pm 1$ |
| (1 week after preparation) | - | - | - |
| Sample B | $194 \pm 6$ | $0.29 \pm 0.06$ | $-58 \pm 3$ |
| (1 week after preparation) | $155 \pm 2$ | $0.41 \pm 0.02$ | $-55 \pm 2$ |
| Sample C | $163 \pm 3$ | $0.30 \pm 0.02$ | $-65 \pm 5$ |
| (1 week after preparation) | $136 \pm 3$ | $0.12 \pm 0.02$ | $-71 \pm 2$ |

The sizes of the particles in these dispersions are much larger than that of the single NPs as determined by XRD analysis. This is an indication that bare NPs and those with adsorbed citrate form aggregates in diluted aqueous dispersions at neutral pH. The aggregates are larger for bare NPs than for the other samples. The effect of citrate adsorption can be seen on the longterm stability of the dispersions. For bare NPs, one week after preparation the intensity of the light scattered by the residual particles in the dispersions was so week that reliable DLS measurements could not be performed. This is the result of the formation of aggregates which grow in size and eventually sediment. For the samples B and C, the aggregation process of the particles present in the dispersions immediately after sonication is prevented by the citrate coating. As a result, one week after preparation, the majority of particles remains in the dispersions. The larger aggregates initially present sedimented and a decrease in the Z-average is observed (Figure 3, dashed lines and Table 3). These results can be explained taking into account the $\zeta$-potential values measured for the samples (Table 3). Due to adsorption of citrate anions, B and C NPs have a $\zeta$-potential which, in absolute value, is significantly larger than that of bare NPs and higher than 30 mV, the limiting value for particle dispersions to be stable against aggregation. Hence, for the citrate covered particles, the electrostatic repulsion between the negatively charged aggregates prevents their adhesion and growth.

Z-average values tend to overestimate the actual size of the particles since larger particles contribute more to the scattered intensity. In fact, the hydrodynamic diameter determined from the fraction number of particles vs. size curves is smaller than that obtained from the intensity versus hydrodynamic diameter curves (Figure S5, Supplementary Materials). However, even taking this effect into account, the size of the particles in the dispersion is much larger than those of the single NPs. Aggregates of magnetite NPs form even when the co-precipitation is carried out in the presence of citrate due to the relatively high ionic strength of the solution. Although the NPs are negatively charged due to the adsorption of citrate anions, the electrostatic repulsions is reduced because of the screening exerted by the ions in solution. Hence, NPs can get close and adhere one another. Since the clusters cannot be disaggregated by sonication it seems that the primary NPs are attached together by interactions stronger than those due to Van der Waals forces or hydrogen bonds. These weaker interactions would lead to agglomerates (rather than aggregates) which could be dissolved by sonication. It is conceivable that magnetite NPs covered by citrate in situ can attach one other sharing surfaces of the nanocrystals not covered by citrate anions. Adsorption of citrate (and other carboxylic acids) occurs specifically on faces delimiting a nanocrystal with given orientations [33]. NPs can join through these uncoated faces, such as in the twinning of crystals. This mechanism is responsible for the oriented attachment leading to "mesocrystals" or to "nanoflowers" of magnetite NPs [34,35]. Thus, the primary NPs of magnetite can be seen as building blocks of more complex structures.

*3.3. Magnetic Properties*

3.3.1. Magnetization Measurements

Magnetization vs. magnetic field intensity curves measured for all the three samples at 300 and 2.5 K are shown in Figure 4a,b and Figure 5. The results of these magnetic measurements are summarized in Table 4. The magnetization curves measured for the three types of NPs at RT do not exhibit any residual magnetization (remanence) when the intensity of the applied magnetic field is zero (Figure 4a). When the magnetization curves are normalized to their respective maximum values (saturation or close to saturation magnetization, $M_s$) the M vs. H curve of sample C tends to the saturation magnetization in a different way compared to the those of the other samples (Figure 4b). This is probably due to the different sizes of the in situ NPs with respect to ex situ and bare NPs. The curves measured at low temperature show a hysteresis loop with a remanence (Figure 5 and Table 4). These results demonstrate that the three types of NPs have a superparamagnetic behaviour. The saturation magnetization of bare NPs is within the range of values reported in the literature for magnetite NPs with a size around 10 nm but less than that of bulk magnetite (92 emu/g). This reduction in $M_s$ for magnetite and other iron oxides in the form of NPs could be due to the "magnetically dead layer" at the surface of the NPs [36]. However, the less ordered crystal structure of the NPs prepared by co-precipitation compared to those obtained by high temperature synthetic routes could be responsible for the lower $M_s$ [37]. The lower $M_s$ may be due also to the formation of a layer of maghemite at the surface of the NPs since bulk maghemite has a $M_s$ lower (74 emu/g) than magnetite. Bare and ex situ coated NPs have comparable $M_s$ and the smaller value of the latter could be tentatively attributed to a reduction in the magnetic moment at the surface of the NPs due to the bonding of citrate ions.

**Table 4.** Saturation magnetization $M_s$ measured at 300 K and 2.5 K, ratio of the residual ($M_r$) and $M_s$ at 2.5 K, coercive field (Oe) at 2.5 K for the three samples.

| Sample | $M_s$ (emu/g) at 300 K | $M_s$ (emu/g) at 2.5 K | $M_r/M_s$ | $H_c$ (Oe) |
|---|---|---|---|---|
| A (bare NPs) | 66 | 74 | 0.31 | 423 |
| B ($Fe_3O_4$ NPs cit. ex situ) | 58 | 68 | 0.29 | 342 |
| C ($Fe_3O_4$ NPs cit. in situ) | 40 | 51 | 0.23 | 511 |

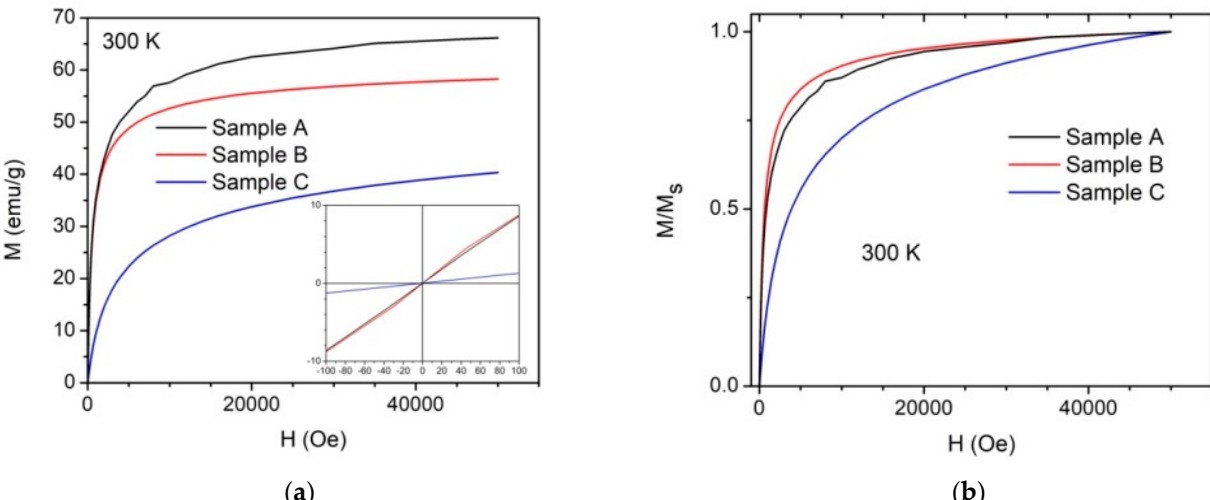

(**a**)    (**b**)

**Figure 4.** Magnetization (emu/g) vs. magnetic field intensity (Oe) curves of samples A, B and C measured at RT. The inset in the low right part of the figure shows that the NPs do not have a remanence at 300 K (**a**); Curves of the magnetization vs. magnetic field intensity normalized to their respective maxima of samples A, B and C measured at RT (**b**).

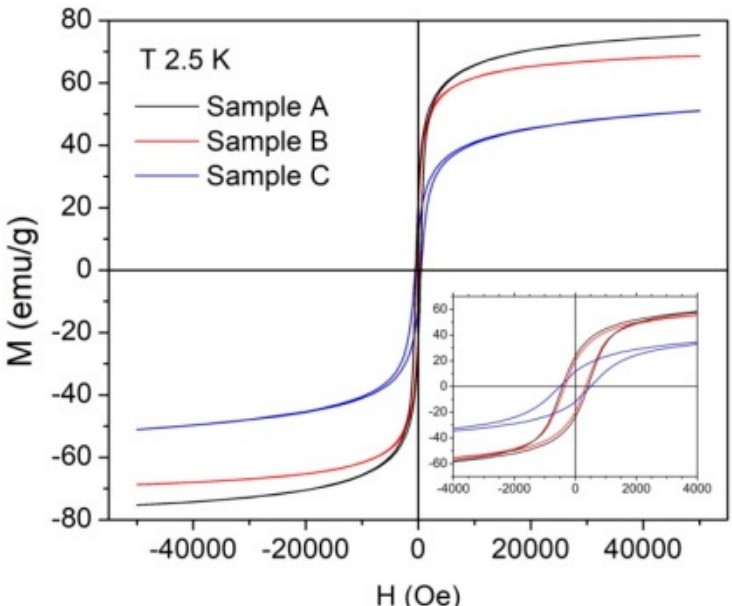

**Figure 5.** Magnetization vs. magnetic field intensity curves of samples A, B and C measured at 2.5 K. In the inset an enlarged region around the zero of the magnetic field intensity is shown.

The $M_s$ for the in situ NPs is much smaller than for the other two kinds of sample and this is consistent with the smaller size of the NPs as determined by XRD. The $M_r/M_s$ for the three samples is ca. 0.3, well below the value (0.5) expected for non-interacting, single domain, uniaxial superparamagnetic NPs. This is an indication of magnetic interactions among the NPs in the aggregates [38]. The coercive fields, comparable to the values reported for magnetite NPs of similar size at temperature around 2–5 K [39], do not show a clear dependence on the particle size. The superparamagnetic behaviour of the NPs is confirmed by the FC/ZFC measurements (Figure 6).

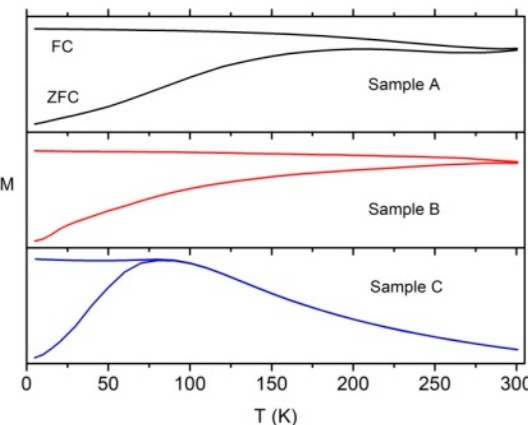

**Figure 6.** ZFC and FC curves measured for the three samples.

The magnetization curves vs. T measured after cooling the sample without an applied magnetic field (ZFC) of bare and ex situ citrate-coatedcitrate-coated NPs do not show a peak to allow the determination of the blocking temperature ($T_B$). Indeed, the ZFC curve of sample A shows a broad peak around 200 K which might correspond to $T_B$. Such broad ZFC curves suggest a polydispersion of the primary NPs and/or their aggregates and the existence of magnetic interactions among the NPs [40]. The ZFC curve of sample C shows a peak at 85 K which corresponds to the $T_b$ of the NPs. $T_b$ is much higher than the value expected for non-interacting magnetite NPs with a diameter of 5 nm. The observed higher $T_b$ can be attributed to the magnetic dipole interactions among the NPs. These interactions and the size distribution of the NPs are responsible for the broadness of the peak. Hence, the ZFC-FC curves can be explained taking into account the smaller size of the in situ citrate-coated magnetite NPs compared to bare and ex situ citrate-coated NPs. Large aggregates of magnetite NPs (size 100–200 nm) should be ferrimagnetic since their size is well above the limiting value (10–30 nm) for single domain, superparamagnetic magnetite particles. On the contrary magnetic measurements do show that all the three samples are superparamagnetic. Ge et al. reported that aggregates of magnetite NPs with a size around 100 nm exhibit a superparamagnetic behavior [41]. It is remarkable that the magnetic properties of the aggregates of magnetite NPs, bare as well as coated with citrate, are dominated by those of the primary NPs. On the other hand for aggregates (40 nm in diameter) of magnetite single NPs (ca. 10 nm in diameter), prepared by co-precipitation in the presence of polyarginine, it was found that the magnetic properties are determined essentially by the size of the aggregates rather than by that of the constituents NPs. This behaviour was attributed to the same crystallographic orientation of the individual NPs [42].

3.3.2. Size of the Magnetic Domains

By analysing magnetization vs. magnetic field intensity curves of superparamagnetic NPs above $T_b$, it is possible to determine the average size of magnetic domains and their size distribution function [28]. If the NPs are single domain, the size of the magnetic domains coincides with that of the single NPs. In the case of non-interacting magnetic particles, the magnetization vs. field intensity curves can be modelled by using this function:

$$M(H) = M_s \int_0^\infty L(H, D) P_V(D) dD \qquad (1)$$

where $D$ is the diameter of the particles, *L(H,D)* is the Langevin function which depends on the magnetic dipole moment (and thus on D) and on the magnetic field intensity and $P_v(D)$ is the volume-weighted diameter distribution function [27,29,43,44]. Equation (1) was used to fit the experimental magnetization vs. magnetic field intensity curves to derive the average size of the magnetic domains and the standard deviation of the size distribution (see the Supplementary Materials). The lognormal distribution function was used for

$P_v(D)$ [27,44]. The experimental magnetization curves and their best fit curves are shown in Figure 7 for all the three samples. For a better comparison the curves are shown on a double log scale.

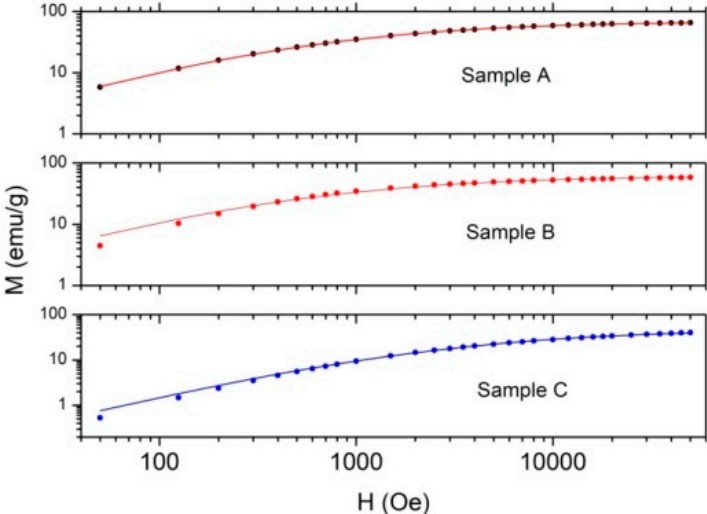

**Figure 7.** Fitting (solid lines) of the experimental magnetization vs. magnetic field intensity curves measured at RT for the three samples (filled circles) by using Equation (1).

The quality of the fit is rather good with only some residual discrepancies in the region of low magnetic field. The results of the analysis, summarized in Table 5, show that the sizes of the magnetic domains are very close to the sizes of the coherence domains as determined by XRD (see Table 2), that is the crystalline and magnetic order have the same extension. The distribution curves of the magnetic domains for the three samples are shown in Figure S6, Supplementary Materials. The curves for samples A and B are broader than that of sample C. However, the relative standard deviations are the same for all the samples (see Table 5).

**Table 5.** Results of the analysis of the magnetization curves for the three samples. $<D_m>$ is the average diameter of the magnetic domains, $\sigma$ the standard deviation of the distribution, $\sigma/<D_m>$ the relative standard deviation. $M_s$ is the saturation magnetization obtained by fitting the experimental magnetization curves.

| Sample | $<D_m>$ (nm) | $\sigma$ (nm) | $\sigma/<D_m>$ | $M_s$ (emu/g) |
|--------|--------------|---------------|----------------|---------------|
| A | $8.7 \pm 0.6$ | $5.1 \pm 0.1$ | $0.6 \pm 0.1$ | $67 \pm 1$ |
| B | $9.9 \pm 0.5$ | $4.9 \pm 0.2$ | $0.5 \pm 0.1$ | $58 \pm 1$ |
| C | $5.1 \pm 0.6$ | $2.8 \pm 0.1$ | $0.5 \pm 0.1$ | $46 \pm 1$ |

Although the NPs are interacting magnetically (as indicated by the ZFC-FC measurements), the good correspondence between the size of the magnetic domains and the size of the individual NPs for these samples shows that magnetization curves can be analysed in terms of Langevin equation, which, on the other hand, consider non-interacting particles. This can be explained taking into account that the relaxation time of the magnetization for superparamagnetic NPs depends exponentially on the effective anisotropy constant which contains an additional contribution due to magnetic interactions among the NPs [40,45]. Hence, even relatively weak interactions are able to produce the observed shift of $T_b$ but do not affect significantly the magnetization curves which can be analysed in the framework of the Langevin model.

### 3.3.3. EPR

EPR measurements were performed to obtain additional information about the effect of size and aggregation of the NPs on their magnetic properties. The EPR spectrum of each type of sample shows only one rather broad peak (Figure 8). These spectra are similar to those reported in the literature for magnetite NPs of similar sizes [7,8,23]. The peak of sample C is narrower and shifted with respect to the peaks of samples A and B, which have similar width and position (Table 6). The shift of the resonance peak can be ascribed to the smaller diameter of NPs C compared to NPs A and B. The magnetic dipoles of NPs generate an internal magnetic field which adds to the applied field [7] thus the intensity of the external field corresponding to the resonance peak is lower. The internal field is larger for sample A (and sample B) than for sample C since the magnetization in the former case is larger than in latter case and therefore the resonance occurs at a lower field.

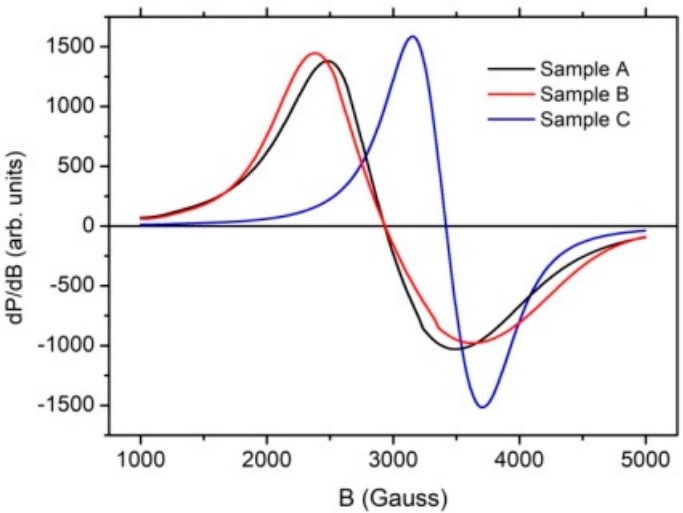

**Figure 8.** X-band EPR spectra measured for samples A, B and C at RT.

**Table 6.** Summary of the parameters derived from the EPR spectra. $B_0$ induction magnetic field corresponding the resonance peak. g factor, $\Delta B$ width of the peak. The asymmetry parameter A is defined as the ratio of the height of maximum to the depth of the minimum of the derivative of the peak.

| Sample | $B_0$ (Gauss) | g | $\Delta B$ (Gauss) | A |
|--------|---------------|---|--------------------|---|
| A | 2932 | $2.41 \pm 0.01$ | 989 | 1.25 |
| B | 2939 | $2.41 \pm 0.01$ | 1160 | 1.27 |
| C | 3417 | $2.06 \pm 0.01$ | 555 | 1.10 |

Both the smaller size and less relevant magnetic dipole–dipole interactions between adjacent NPs can explain the narrower peak width of sample C compared to samples A and B [46]. Hence, EPR results are consistent with the ZFC-FC curves showing a higher Tb of sample A and B compared to C due to their larger size and larger magnetic dipole–dipole interactions. The resonance peaks are slightly asymmetric with a height of the maximum larger than the height of the minimum. The asymmetry ratio (defined as the ratio of the maximum to the minimum) is larger than 1 for all the samples. This is consistent with the negative cubic magnetocrystalline anisotropy [8].

### 4. Conclusions

The results of the present work show that primary magnetite NPs prepared in the presence citrate during the co-precipitation have a significantly smaller size than those prepared by adsorbing citrate post-synthesis. The latter type of NPs have a size which is not significantly different from that of bare NPs. Since all the three kinds of sample have almost

the same relative standard deviations of the PSD, we can conclude that the presence of citrate during the co-precipitation does not influence the level of polydispersion of the NPs. The size of the $Fe_3O_4$ NPs determined by analysing the XRD measurements with different approaches are in good agreement with the magnetometric results. This result confirms the soundness of the hypotheses at the basis of the analysis of the magnetization curves. In particular that the magnetic dipole interactions among the NPs are relatively weak so that the magnetization vs. magnetic field intensity can be modelled by the Langevin equation. The size of the particles in aqueous solutions as determined by DLS are much larger than those of individual NPs indicating the formation of aggregates. In situ coating with citrate does not substantially limit the aggregation of the NPs. Magnetite NPs co-precipitated in the presence of citrate attach one another probably sharing faces of the nanocrystals not covered by citrate anions. Aggregates of citrate-coated NPs, prepared both in situ and ex situ, are smaller than those of bare magnetite NPs. Moreover, their dispersions are colloidally stable due to their negative surface charge imparted by citrate anions which prevents the further growth and sedimentation of the aggregates. Magnetization vs. magnetic field intensity and FC/ZFC curves clearly show that all the three types of sample have a superparamagnetic behaviour, with no residual magnetization at RT. The values of the blocking temperatures and the EPR spectra indicate the existence of magnetic interactions among the individual NPs inside the aggregates. Nonetheless, the magnetic properties of these assemblies are dominated by those of the individual NPs which have a size below that required to be superparamagnetic.

This study shows that by adding citrate during the co-precipitation of magnetite NPs it is possible to control the size of the primary NPs and prevent, at least in part, their aggregation. Since the size of individual NPs and their aggregation have a large impact on their magnetic properties and dispersion stability, the results of this study which concern basic aspects of particle growth and aggregation are relevant for the vast range of applications of these NPs.

**Supplementary Materials:** The following are available online at https://www.mdpi.com/article/10.3390/app11156974/s1, File with Figures S1–S6 and details about the determination of the domain size by analyzing the magnetization vs. magnetic field intensity curves.

**Author Contributions:** Draft preparation, writing, review and editing, A.A., F.F.M. and M.C.B. XRD measurements, A.S. DLS measurements, A.A. and F.F.M. EPR measurements, M.C.B. All authors have read and agreed to the published version of the manuscript.

**Funding:** This research was funded by MIUR (Ministero dell'Istruzione, dell'Universita e della Ricerca), Grant: "Dipartimento di Eccellenza 2018–2022".

**Institutional Review Board Statement:** Not applicable.

**Informed Consent Statement:** Not applicable.

**Data Availability Statement:** Not applicable.

**Acknowledgments:** The authors thank Roman Pielaszek, Pielaszek Research, Poland and Matteo Leoni, Dipartimento di Ingegneria, Civile, Ambientale e Meccanica, Università di Trento, Italy, for the useful discussions and the help with the XRD analysis. The authors are indebted with Claudia Innocenti, Dipartimento di Chimica, Università di Firenze and INSTM, Firenze, Italy, for the magnetic measurements.

**Conflicts of Interest:** The authors declare no conflict of interest.

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
