# Peer review of "Effect of Citrate on the Size and the Magnetic Properties of Primary Fe3O4 Nanoparticles and Their Aggregates"

_applsci, doi:10.3390/app11156974_

Round 1

Reviewer 1 Report

Report of Manuscript applsci-1318117 for Applied Sciences

Title: Effect of citrate on the size and the magnetic properties of primary Fe3O4 nanoparticles and their aggregates by Andrea Atrei et al.

This work focus on the characterization of "superparamagnetic Iron-Oxide-nanoparticles" at different citrate concentrations and through different complementary techniques.

The paper is well-written, the English is very good and I think that this work could be of interest for the field of basic and applicative research of NPs. The paper is technically sound, but it is limited and specific interest. The work is well structured and the proposed goals were achieved. The quality and quantity of the results presented are adequate with the standards of "Applied Sciences"

The manuscript contains new information to justify publication. The methods described comprehensively. The list of references should be improved and modified. The interpretations and conclusions justified by the results. Different and complementary analyzes were carried out. The manuscript appears complete in all its parts.

However, the manuscript should be improved and it will be worth for publication after some minor revisions as recommended below.

Abstract

I suggest that authors avoid acronyms in the abstract.

Introduction

About magnetic nanoparticles, I suggest increasing these references. This research field is very vast and of great scientific interest. The authors must increase references. The introduction should be enriched with articles related to the applications of nanoparticles. I suggest for example to add “https://doi.org/10.3390/nano10101919”; “https://doi.org/10.3390/app10207322” https://doi.org/10.3390/nano10112310 and many others.

Sections 2.1

-The authors should indicate the purity of the reagents used.

-How was the pH evaluated?

Sections 2.5

-How were the parameters of the ESR acquisitions chosen?

Results

- Increase all figures in size and font size.

Conclusions - The conclusions need to be enriched to emphasize the applicability of the results found: this aspect is fundamental to the publication and impact of this manuscript.

-Different minor typo-corrections that should be performed.

Reviewer 2 Report

There are many typos and grammatical mistakes. Therefore, you should carefully read the manuscript again and correct them.

The sentences at the end of Figure 1 caption do not make any sense. (This is a figure. Schemes follow the same formatting.) Please remove them.

σ in Table 1 should be explained in both main text and caption.

What are the vertical axes of Figures 1 and 2?

Supplementary information is missing. The reviewer cannot properly evaluate your manuscript in the present form. If you attached the supplementary information when you submitted it, please complain to the editor.

Round 2

Reviewer 2 Report

This paper can be published after the following correction.

line 121: Fourier Transform InfraRed-->Fourier Transform Infrared

line 197 (Figure caption): smaple C( citrate coated... --> smaple C (citrate coated...

line 268: Why is the Table 2 cation italic? Does it have any special meaning?

Author Response

We corrected the text according to the comment of the reviewer

line 121: Fourier Transform InfraRed-->Fourier Transform Infrared
Corrected

line 197 (Figure caption): smaple C( citrate coated... --> smaple C (citrate coated...
Corrected

line 268: Why is the Table 2 cation italic? Does it have any special meaning?
No special meaning. It was just a mistake in formatting the text. Now the text is not in italic.